# Oral Administration of Liquiritigenin Confers Protection from Atopic Dermatitis through the Inhibition of T Cell Activation

**DOI:** 10.3390/biom10050786

**Published:** 2020-05-19

**Authors:** Hyun-Su Lee, Eun-Nam Kim, Gil-Saeng Jeong

**Affiliations:** College of Pharmacy, Keimyung University, Daegu 42601, Korea; hyunsu.lee@kmu.ac.kr (H.-S.L.); enkimpharm@gmail.com (E.-N.K.)

**Keywords:** liquiritigenin, T cell activation, CD69, MAPK, atopic dermatitis, lymph nodes

## Abstract

While liquiritigenin, isolated from *Spatholobus suberectus Dunn*, is known to possess anti-inflammatory activities, it still remains to be known whether liquiritigenin has a suppressive effect on T cell activation and T cell-mediated disease. Here, we used Jurkat T cells to explore an underlying mechanism of pre-treatment with liquiritigenin in activated T cell in vitro and used atopic dermatitis (AD) in vivo to confirm it. We found liquiritigenin blocks IL-2 and CD69 expression from activated T cells by PMA/A23187 or anti-CD3/CD28 antibodies. The expressions of surface molecules, including CD40L and CD25, were also reduced in activated T cells pre-treated with liquiritigenin. Western blot analysis indicated repressive effects by liquiritigenin are involved in NFκB and MAPK pathways. To assess the effects of liquiritigenin in vivo, an AD model was applied as T cell-mediated disease. Oral administration of liquiritigenin attenuates AD manifestations, including ear thickness, IgE level, and thicknesses of dermis and epidermis. Systemic protections by liquiritigenin were observed to be declined in size and weight of draining lymph nodes (dLNs) and expressions of effector cytokines from CD4^+^ T cells in dLNs. These results suggest liquiritigenin has an anti-atopic effect via control of T cell activation and exhibits therapeutic potential for T cell-mediated disorders.

## 1. Introduction

*Spatholobus suberectus* Dunn (*S. suberectus*), a traditional Korean medicinal plant, has shown several pharmacological activities that include anti-inflammatory, anti-fungal, anti-bacterial, and anti-tumor activities [1,2,3]. Various compounds have been reported as components of *S. suberectus*, including formononetin, prunetin, plathymenin, calycosin, butin, eriodictyol, dihydroquercetin, dihydrokaempferol, and liquiritigenin [4]. Liquiritigenin (4′,7-dihydroxyflavanone, C_15_H_12_O_4_) has been mainly isolated from the root of Glycyrrhiza uralensis (licorice) [5] but has also been discovered in several plants, such as *Sinofranchetia chinensis* [6], *Jacaranda obtusifolia* [7], and *Pterocarpus marsupium* [8]. Liquiritigenin has been found to possess anti-oxidative [9], anti-tumor [10], and anti-cardiotoxicity [11] activities. Accumulating evidence has demonstrated the effect of liquiritigenin on inflammatory condition targeting liver injury [12], cartilage matrix degeneration [13], and hepatic injury [14]; however, few studies have reported that liquiritigenin has inhibitory effects on T cell activation and T cell-mediated inflammation.

When naïve helper T cells are shown with antigenic peptides by MHCII molecules, which are expressed on antigen presenting cells (APCs), including dendritic cells, B cells, and macrophages, T cell receptor (TCR) signal pathway triggers initial activation of T cells, including the expression of surface molecules [15]. Up-regulated expression of CD69 acts as a co-stimulatory molecule and leads T cells proliferation [16]. IL-2 is also produced in the early step of T cell activation that serves T cell proliferation and differentiation [17]. Since CD69 and IL-2 are expressed in the first step of T cell activation, they have been considered as markers of T cell activation [18,19]. After the encounter of naïve T cells with APCs, naïve T cells can be differentiated to effector T cells that induce the expression of CD40L and CD25 (IL-2 receptor) for B cell helpers and proliferation [20,21]. The NFκB pathway, including p65 translocation and MAPK pathway, are known to be involved in T cell activation. Understanding the process of T cell activation is critical for developing novel therapeutics of T cell-mediated diseases including atopic dermatitis (AD).

AD is one of the multi-factorial diseases that is caused by environmental or genetic issues; hence, it is considered an incurable disease [22]. During recent decades, although many therapeutic approaches to conquer AD have been tried by understanding the mechanism of AD development, few trials have demonstrated the importance of T cells in AD. Once naïve T cells are primed and activated by dendritic cells that load allergen peptides, they differentiate into effector T cells in lymph nodes to lead pathogenesis by producing effector cytokines, including IL-4, IL-5, IL-6, and IL-13 [23,24]. With Th2 cytokines milieu from effector T cells, AD is developed, and severe inflammatory response is generated. As mentioned above, T cells play a critical role in AD progress, so that regulation of T cell activation is a promising strategy for improving AD symptoms [25,26]. However, it is still unknown whether treatment with liquiritigenin abrogates T cell activation in vitro and protects from atopic dermatitis in vivo. Here, we explored the effect of liquiritigenin isolated from *S. suberectus* on T cell activation with underlying mechanism and therapeutic potential of oral administration of liquiritigenin for AD pathogenesis.

## 2. Materials and Methods

### 2.1. Cell Culture

Jurkat T cells were purchased from Korean Cell Line Bank (Seoul, Republic of Korea). The cells were cultured in RPMI medium (Welgene, Gyeongsan-si, Republic of Korea) supplemented with 10% fetal bovine serum (FBS), penicillin G (100 units/mL), streptomycin (100 μg/mL), and L-glutamine (2 mM), and grown at 37 °C in a humidified incubator containing 5% CO_2_ and 95% air.

### 2.2. Mice

Eight-week-old female BALB/c mice were obtained from Samtako and housed in specific pathogen-free (SPF) conditions. All experiments were approved by the Animal Care and Use Committee of the College of Pharmacy, Keimyung University (approval number: KM2019-005).

### 2.3. Plant Materials

The dried of *S. suberectus* was purchased from the Yangnyeong herbal medicine market (Daegu, Korea, in June 2019). A voucher specimen (KMU-2019-11-16) of the plant was deposited at the College of Pharmacy in Keimyung University.

### 2.4. Extraction and Isolation

The dried stem of *S. suberectus* (10 kg) was refluxed with 100% ethanol for 3 h at boiling temperature. The dried EtOH (1.72 kg) extract was suspended with H_2_O, and the resulting H_2_O layer was partitioned three times with hexane (486 g), EtOAc (841 g), and H_2_O (393 g). The EtOAc-soluble fraction was loaded onto silica column (8 × 60 cm, silica-gel 70-230mesh), eluted in methanol in H_2_O (gradient from 0:100 to 100:0) to obtain seven fractions (Fr.1 to Fr.10). Among them, Fr.5 was subjected to Sephadex LH-20 column chromatography (35% MeOH to 100% MeOH) to obtain 8 fractions (Fr.5-1 to Fr.5-8). The Fr.5-8 was performed to C18 column chromatography followed by elution with a gradient solvent system of MeOH in H_2_O (45% MeOH to 100% MeOH) and purification with a semi-preparative high-performance liquid chromatography (HPLC) to giving liquiritigenin (274 mg). Isolated liquiritigenin was identified by comparing the values of spectroscopy data from previously published literature [27]. The isolated liquiritigenin was detected at 35.7 min with purity of 94% (Figure 1A, top), and liquiritigenin in EtOAc fraction of *S. suberectus* was also detected at 35.7min (Figure 1A, middle) but not in the hexane fraction (Figure 1A, bottom). The structure of liquiritigenin is shown in Figure 1B.

### 2.5. Condition of High-Performance Liquid Chromatography (HPLC) Analysis

Analyses were performed using a reversed-phase high-performance liquid chromatography (HPLC) system (Shimadzu, Japan) with a Luna C18 column (5 μm × 4.6mm × 150 mm; Phenomenex, USA) and diode array detector. The analysis was performed for 55 min by injecting 10 μL of the sample at a flow rate of 1 mL/min. The mobile phase consisted of 0.1% formic acid water (A) and methanol (B) in a ratio specified by the following binary gradient with linear interpolation: 0 min 15% B, 30 min 85% B, 40 min 85% B, and 55 min 15% B.

### 2.6. Reagent and Antibodies

Anti-phosphorylated mTOR (S2448) antibodies and anti-Bcl2 were purchased from Santa Cruz Biotechnology (Dallas, TX, USA). Anti-p65, anti-PARP, anti-IκBα, anti-phosphorylated IκBα (S32), anti-β-actin, anti-phosphorylated ERK (T202/Y204), anti-ERK, anti-phosphorylated p38 (T180/Y182), anti-p38, anti-phosphorylated JNK (T183/Y185), anti-JNK, anti-phosphorylated c-Jun (S63), and anti-c-Jun were obtained from Cell Signaling Technology (Danvers, MA, USA). MTT (1-(4,5-Dimethylthiazol-2-yl)-3,5-diphenylformazan), phorbol 12-myristate 13-acetate (PMA), A23187, and dinitrochlorbenzene (DNCB) were purchased from Sigma Chemical Co. (St. Louis, MO, USA). Anti-human CD3 and CD28 antibodies were obtained from Bioxcell (West Lebanon, NH, USA). Human IL-2 ELISA kit was purchased from R&D systems (Minneapolis, MN, USA). AnnexinV staining dye and caspase3/7 staining dye for IncuCyte^®^ cell imaging system were obtained from Essen bio (Ann Arbor, MI, USA). Antibodies to CD69, CD25, and CD40L were purchased from ebiosciences (San Diego, CA, USA). NE-PER Nuclear and Cytoplasmic Extraction Reagents Kit was obtained from Thermo Fisher Scientific (Waltham, MA, USA). Mouse IgE ELISA kit was purchased from BD Biosciences (San Diego, CA, USA). House dust mite extract was obtained from Greer (Lenoir, NC, USA).

### 2.7. MTT Assay

Cell viability was measured using the MTT assay. Jurkat T cells (1 × 10^4^/well in 96-well plate) were treated with the indicated concentration of (0–40) μM of liquiritigenin for 24 h at 37 °C. MTT (500 μg/mL) was added to cells for 1 h. Cell culture plates were centrifuged, supernatant of each well was discarded, and 150 μL of DMSO was added to dissolve formazan crystals. Plates were read at 540 nm, and the absorbance values of each sample were measured. Cell viability was calculated with the absorbance value of samples and control (% of control).

### 2.8. T Cell Stimulation

Jurkat T cells (1 × 10^6^/well in 12-well plate) were pre-treated in the presence or absence of the indicated concentration of liquiritigenin for 1 h. For PMA/A23187 stimulation, cells were incubated with 100 nM of PMA and 1 μM of A23187 for the indicated time. For TCR-mediated stimulation, cells were loaded on the plate pre-coated with 10 μg/mL of anti-CD3 antibodies, and 3 μg/mL of anti-CD28 antibodies was added for the indicated time.

### 2.9. ELISA

For human IL-2 measurement, activated Jurkat T cells were discarded, and supernatants were collected. Collected supernatants were used for human IL-2 measurement. For IgE determination, blood samples were removed from mice, and centrifuged at 7000 rpm for 5 min. After centrifugation, isolated serum level was collected for further experiments. ELISA was performed by following the instructions of the kit manufacturer.

### 2.10. Flow Cytometry

Jurkat T cells (5 × 10^5^/well in 24-well plate) were stimulated with PMA/A23187 for 16 h and harvested for staining. Cells were stained with antibodies against the indicated antigens for 1 h at 4 °C. After washing with cold PBS, cells were acquired by flow cytometry (BD FACSVerse). The histograms were generated using BD software.

### 2.11. Western Blot Analysis

Stimulated Jurkat T cells were lysed in radioimmunoprecipitation assay (RIPA) buffer (1% Triton X-100, 150 mM NaCl, 20 mM Tris pH 7.5) with protease inhibitor and phosphatase inhibitor for 30 min at 4 °C. After centrifugation at 14,000 rpm for 20 min at 4 °C, supernatants were collected, and concentrations were measured for loading. Approximately (30 to 50) μg of the lysate was loaded on (8–12)% SDS–PAGE gels. Separated proteins were transferred onto PVDF membrane (Bio-Rad, Hercules, CA, USA). The membrane was blocked in blocking buffer (5% skim milk) for 1 h, rinsed, and incubated with the indicated primary antibodies in 3% skim milk overnight at 4 °C. Excess primary antibodies were washed out in TBS with 0.1% of Tween 20 (TBS-T) three times and then incubated with 0.1 μg/mL peroxidase-labeled secondary antibodies (against mouse or rabbit) for 2 h. After three washes with TBS-T, bands were visualized with ECL Western blotting detection reagents with ImageQuant LAS 4000 (GE Healthcare, Chicago, IL, USA). All observed bands were quantified with ImageJ software and normalized with control.

### 2.12. NFκB Translocation Analysis by Western Blot

For investigation of p65 translocation, NE-PER Nuclear and Cytoplasmic Extraction Reagent was used to collect nucleus extracts, following the manufacturer’s instruction. Stimulated Jurkat T cells in different condition were lysed in a CER buffer for 10 min at 4 °C, and centrifuged at 14,000 rpm for 5 min, to isolate cytosolic extracts from whole lysates. After centrifugation, supernatants were clearly removed for cytosol extract, and pellets were lysed in NER buffer for an additional 40 min at 4 °C. After centrifugation, supernatants were harvested as nucleus proteins. For normalization of each sample from nucleus proteins and cytosol proteins, anti-PARP and anti-β-actin antibodies, respectively, were used as loading control.

### 2.13. AD induction in Mice Ear

AD in BALB/c mice was developed by alternately repeated topical administration of mite extract and DNCB on mice ears, as previously described [28]. There were four mice groups, namely, healthy untouched mice that were not administrated with either DNCB/mite extract or liquiritigenin (con), mice that were administrated with liquiritigenin alone (LG), control mice that were treated with DNCB/mite extract alone (AD), and experimental mice that were treated simultaneously with both DNCB/mite extract and liquiritigenin (AD + LG). Each group had six mice in the same cage. For induction of AD, surfaces of both ears were stripped five times with surgical tape (Seo-il Chemistry, Hwasung, Korea). After stripping, each ear was topically administrated with 20 µL of DNCB (1%). Four days later, ears were painted with 20 µL of mite extract (10 mg/mL). The mite extract/DNCB administration was weekly repeated for 4 weeks. After a 2-day break, this 5 days-on and 2 days-off oral administration of liquiritigenin was repeated for 4 weeks (total 20 times). Ear thickness was measured at 24 h after administration of DNCB/mite extract, by using a dial thickness gauge (Kori Seiki MFG Co., Tokyo, Japan). Animals were sacrificed on day 28.

### 2.14. Histological Analysis

After sacrifice, ears were removed, and used for histopathological analysis. These collected ears were fixed in 10% paraformaldehyde and embedded in paraffin. The paraffin-embedded tissues were sliced into 5 µm thick sections, deparaffinized, and stained with hematoxylin and eosin (H&E). H&E-stained slides were used to measure the thickness of the dermis and epidermis.

### 2.15. Isolation of CD4^+^ T Cells

After sacrifice, dLNs were collected from each mouse. CD4^+^ T cells were separated from dLNs by magnetic-activated cell sorting (MACS) separation (Miltenyi Biotec, Bergisch Gladbach, Germany).

### 2.16. Statistics

Mean values were calculated from the data obtained from at least three separate experiments performed on separate days. The significance between multiple groups was determined by one-way ANOVA test. Differences between groups were considered significant at *p* < 0.05.

## 3. Results

### 3.1. Liquiritigenin Does not Induce Cellular Apoptosis

We first evaluated whether liquiritigenin shows cytotoxicity in Jurkat T cells during culture. Jurkat T cells were treated with the indicated concentration of liquiritigenin for 24 h, and cell images were collected by IncuCyte^®^ cell imaging system. Cell confluency of each group showed comparable levels from all groups (Figure 2A, top panel). To explore whether apoptosis is involved in the treatment with liquiritigenin, the expressions of AnnexinV and caspase3/7 from incubated cells were measured. The expressed AnnexinV and caspase 3/7 were approximate in all conditions (Figure 2A, middle and bottom panels). To confirm the cytotoxicity of liquiritigenin, MTT assay was performed. Figure 2B shows that treatment with liquiritigenin of (0 to 40) µM was not cytotoxic to Jurkat T cells. These results indicate that liquiritigenin does not affect cellular cytotoxicity, including apoptosis.

### 3.2. Treatment of Jurkat T Cells with Liquiritigenin Blocks T Cell Activation

IL-2 plays a critical role in T cell proliferation, differentiation, and activation, and has been one of the early activation markers of T cells [17]. To investigate whether liquiritigenin possesses an inhibitory effect on T cell activation, mRNA level of *il2* was monitored from activated Jurkat T cells with PMA/A23187 (Figure 3A) or anti-CD3/CD28 antibodies (Figure 3B). Experiments in dose-dependent or time-dependent manner showed that liquiritigenin effectively declined the expression of *il2* on mRNA level in Jurkat T cells treated with liquiritigenin (Figure 3A,B). Produced level of IL-2 from Jurkat T cells stimulated with PMA/A23187 or anti-CD3/CD28 antibodies was consistently mitigated in the presence of liquiritigenin (Figure 3C). We also confirm the expression of CD69 that is expressed on the surface in activated T cells. Figure 3D shows that expressed CD69 was reduced in activated Jurkat T cells pre-treated with liquiritigenin. The results of Figure 3 suggest that T cell activation is effectively blocked by pre-treatment with liquiritigenin in a dose- or time-dependent manner.

### 3.3. Liquiritigenin Reduces the Expression of Surface Molecules on Activated T Cells

Since the encounter of naïve T cells with APCs leads the expression of other surface molecules including CD40L and CD25 for humoral immunity or T cell proliferation, we investigated whether liquiritigenin affects the expressions of these molecules after T cell stimulation by PMA/A23187 or anti-CD3/CD28 antibodies. Flow cytometry analysis revealed that pre-treatment with liquiritigenin of activated T cells impeded the expression of CD40L which is ligand of CD40 expressed on APCs (Figure 4A). The expression of CD25, which is known as IL-2 receptor, was also explored in activated T cells by PMA/A23187 or antibodies against CD3 and CD28. Figure 4B shows that the expression of CD25 was diminished in activated Jurkat T cells pre-treated with liquiritigenin, compared to control. These data imply that liquiritigenin not only regulates IL-2 and CD69, but also controls surface molecules that are involved in late phase of T cell activation.

### 3.4. Liquiritigenin Inhibits p65 Translocation and MAPK Signaling Pathways in Activated T Cells

NFκB pathway is considered to be one of the pivotal pathways for regulating T cell activation [29,30,31]. To explore whether liquiritigenin had an inhibitory effect on T cell activation involved in NFκB pathway, p65 translocation from cytoplasm to nucleus was measured by Western blot analysis. Shifted p65 to nucleus by PMA/A23187 stimulation was significantly decreased by pre-treatment with liquiritigenin (Figure 5A). Elevated level of phosphorylated IκBα was diminished in Jurkat T cells pre-treated with liquiritigenin. In the presence of liquiritigenin, IκBα degradation was also blocked (Figure 5A). To evaluate the downstream signaling pathway of NFκB affected by liquiritigenin treatment in activated T cells, MAPK pathways were monitored by Western blot analysis. Figure 5B shows that phosphorylated ERK, p38, JNK, and c-Jun were dramatically declined from PMA/A23187-stimulated Jurkat T cells in the presence of liquiritigenin. These results imply that down-regulated T cell activation by pre-treatment with liquiritigenin occurs through NFκB and MAPK pathways.

### 3.5. Oral Administration of Liquiritigenin Mitigates Atopic Dermatitis in Mice

Since AD is one of the systemically T cell-mediated disorders in which T cell activation and differentiation into Th2 effector T cells are important [25,26], we applied the inhibitory effect of liquiritigenin on T cell activation to an AD model in vivo. AD was induced by alternate repeated painting of DNCB and mite extract for 5 weeks, as shown in Figure 6A. Liquiritigenin was orally administrated to untouched mice (LG) as control and AD mice (AD + LG). After 4 weeks post induction, AD was dramatically developed, so that ears of AD mice were swollen and red; however, AD with liquiritigenin mice exhibited improved manifestations (Figure 6B). Compared to control mice, ear thickness was also reduced in AD with liquiritigenin mice on day 8, and at all time points thereafter (Figure 6C). To determine whether oral administration of liquiritigenin attenuates tissue inflammation in ear, histological analysis was performed. H&E staining results exhibited that changes of tissue structure by AD induction were highly recovered by liquiritigenin administration (Figure 6D). In particular, thicknesses of dermis and epidermis were dramatically declined in the group of AD mice with liquiritigenin (Figure 6E). To confirm the systemic improvement in the group of AD mice with liquiritigenin, IgE level in serum was monitored. As expected, IgE level from AD mice administered with liquiritigenin revealed down-regulation, compared to IgE level from the AD mice group (Figure 6F). Therefore, these data suggest that oral administration of liquiritigenin attenuates AD manifestations in mice.

### 3.6. Oral Administration of Liquiritigenin on AD Mice Systemically Decreases the Expression of Effector Cytokines from Effector T Cells

Previous reports have demonstrated that oral administration of immunoregulatory compounds to AD mice leads to an alleviation of systemic immunological responses during AD progress [28,32]. Since we described that liquiritigenin has an inhibitory function on T cell activation in vitro, we investigated whether oral administration of liquiritigenin on AD mice blocks T cell activation, including the production of effector cytokines. Photos of non-draining lymph nodes (non-dLNs) and draining lymph nodes (dLNs) were obtained on sacrifice day and indicated that dLNs of AD mice were highly enlarged compared to dLNs of AD mice with liquiritigenin; however, non-dLNs showed no change from each group (Figure 7A). Figure 7B revealed not only the size of dLNs of AD mice with liquiritigenin was diminished compared to that of the dLNs of AD mice, but weight was also consistently reduced. To explore whether the expressions of effector cytokines were affected by the administration of liquiritigenin in AD, mRNA level of cytokines were checked from CD4^+^ T cells isolated from dLNs. Quantitative PCR results exhibited that effector cytokines exacerbating AD, including IL-4, IL-5, IL-13, IL-31, TNFα, and IL-17, were significantly decreased in group of AD mice with liquiritigenin (Figure 7C). These results imply that oral administration of liquiritigenin on AD mice has systemic influence on the expression of effector cytokines from activated T cells in dLNs.

## 4. Discussion

The present study explored the protective effect of liquiritigenin on AD with underlying mechanism, indicating that it effectively blocks T cell activation without cytotoxicity. Pre-treatment with liquiritigenin of Jurkat T cells reduces IL-2 production as well as CD69 expression on surface from stimulated cells with PMA/A23187 or anti-CD3/CD28 antibodies via NFκB and MAPK pathways. Furthermore, liquiritigenin mitigated the expression of surface molecules, which play a crucial role in late phase of T cell activation. Oral administration of liquiritigenin revealed systemically improved manifestation of AD, which is one of the T cell-mediated disorders by ameliorating the expression of effector cytokines from CD4^+^ T cells in vivo.

Our data revealed that the size and weight of dLNs from AD mice were increased, compared to those of control mice. In addition, expressions of effector cytokines that play a pivotal role for AD pathogenesis were significantly elevated in CD4^+^ T cells from the dLN of AD mice. Previous report demonstrated that enhanced T cell trafficking and proliferation in dLNs were involved in the enlarged size and weight of dLNs [28]. Our data exhibited that oral administration of liquiritigenin attenuates swelling of dLNs in AD mice, and expression of effector cytokines from CD4^+^ T cells (Figure 7). It has been proven that the migrating capacity of T cells is dependent on T cell activation [33] and that T cell activation is a required process leading to T cell proliferation [34]. In the present study, in vitro results described that T cell activation was regulated by pre-treatment with liquiritigenin (Figure 2). Furthermore, accumulating evidence explained the involvement of liquiritigenin on PI3K/Akt pathway, suggesting that it possesses anti-tumor properties by inhibiting cell proliferation and migration via PI3K/Akt pathway in human lung adenocarcinoma [35]. Our data assumed that liquiritigenin might also negatively affect T cell migration into dLNs in AD condition through the downregulation of PI3K/Akt pathways. Further studies should state the effect of liquiritigenin on T cell migration via PI3K/Akt pathways.

Naïve T cell activation is initiated by forming tight conjugation with APCs termed immunological synapse (IS) [36]. The involvement of several co-stimulatory molecules or adhesion molecules has been studied in IS formation, including CD40L. CD40L, or CD154, is known as a ligand of CD40 expressed on APCs, and its engagement between CD40L and CD40 plays a pivotal role in adaptive immunity [20]. Since it has been investigated that CD40L is constitutively expressed on T cells, CD40L-CD40 ligation might play a role as an adhesion molecule in IS in the early phase. After the encounter of T cells with APCs, this interaction not only acts on triggering full activation of T cells in inflammatory condition, but it also enhances the activation of APCs, including B cell differentiation, macrophage activation, and the production of pro-inflammatory cytokines by dendritic cells [20,37,38]. In particular, signaling through CD40 has been known as critical for antibody production of activated B cells [20]. In the late phase of T cell activation, proliferation of T cells might be induced by IL-2 which is produced from neighbor T cells [39,40]. To absorb more IL-2, CD25 (IL-2 receptor alpha) can be gradually induced in the course of TCR-mediated activation. In this study, we found that pre-treatment with liquiritigenin significantly downregulated induction of CD40L and CD25 expression in activated T cells (Figure 4). These results indicate that attenuation of T cell activation by liquiritigenin in T cells hinders CD40L expression that affects the function of APCs, as well as declines CD25 expression to not proliferate T cells in late phase of T cell activation.

The initial step for AD development is priming naïve CD4^+^ T cells by Langerhans cells loaded with allergenic peptides. TCR-mediated signaling pathways lead naïve CD4^+^ T cells to be activated and differentiated into Th2 effector T cells with milieu of Th2 cytokines. IL-4, key cytokines of Th2 effector T cells, and IL-13 provoke B lymphocytes to produce IgE, and the IgE level in serum is dramatically enhanced. Once released, IgE binds to mast cells, which are known as a professional histamine producer, and a tremendous amount of histamines is generated from primed mast cells, and leads to severe pruritus, which is the most common manifestation of AD [41,42,43,44]. Our ELISA data exhibited that IgE level from blood serum of AD mice with liquiritigenin was significantly declined compared to that from AD mice (Figure 6F). This result implies oral treatment with liquiritigenin consistently regulates B cells to generate IgE by restrain of IL-4 and IL-13 expression from activated Th 2 cells. Liquiritigenin not only intrinsically suppresses T cell activation, but also extrinsically influences overall immune responses, including B cell activation and IgE generation. This expected mechanism clearly explains why controlling T cell activation is an attractive target in developing therapeutics for T cell mediated disorders. In the present study, we assessed the potentials of liquiritigenin as a cure for AD for the first time.

## Figures and Tables

**Figure 1 biomolecules-10-00786-f001:**
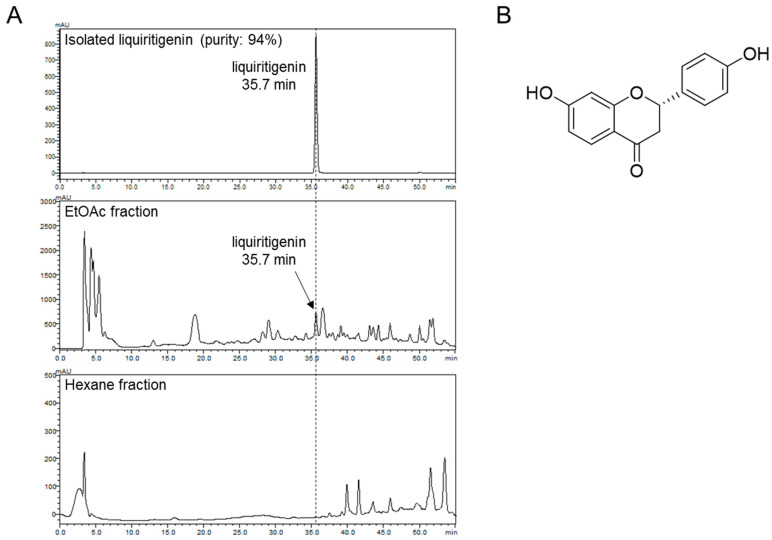
Liquiritigenin is isolated form EtOAc fraction of S. suberectus. (**A**) High-performance liquid chromatography (HPLC) chromatograms of isolated liquiritigenin (top), EtOAc fraction of *S. suberectus* (middle), and n-hexane fraction of *S. suberectus* at 280 nm. (**B**) Chemical structure of liquiritigenin.

**Figure 2 biomolecules-10-00786-f002:**
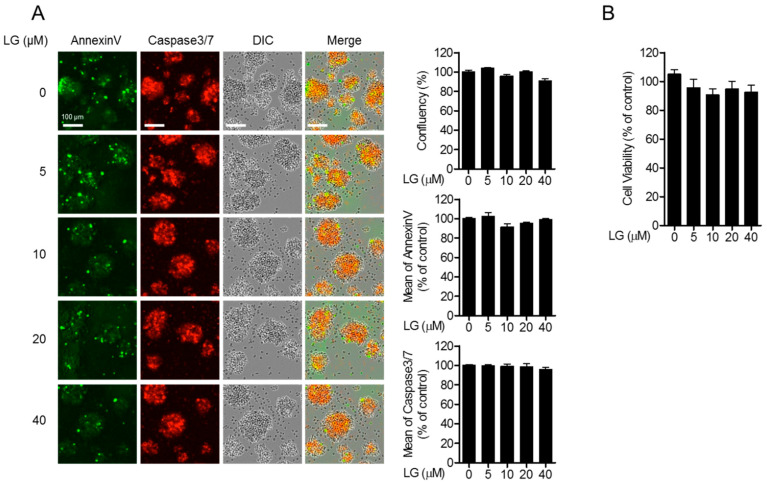
Liquiritigenin does not induce cellular apoptosis. (**A**) Jurkat T cells (5 × 10^4^/well in 96-well plate) were seeded in 96-well plate and treated with the indicated concentration of liquiritigenin of (0 to 40) μM) for 24 h. The confluency of Jurkat T cells was assessed by differential interference contrast (DIC) images obtained by IncuCyte^®^ cell imaging system (top). The mean fluorescence intensities of AnnexinV and caspase3/7 were determined by IncuCyte^®^ cell imaging system (middle and bottom). (**B**) Cell viability was measured by MTT assay. White bars in the micrograph panels represent 100 μm. All results are presented after normalization with the % of control.

**Figure 3 biomolecules-10-00786-f003:**
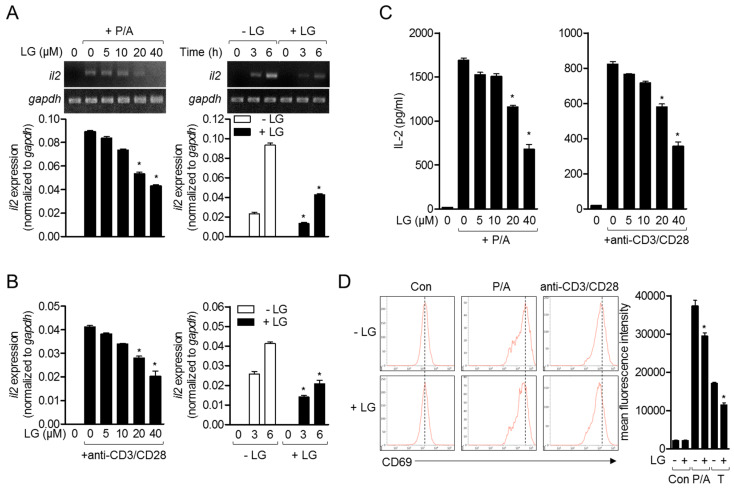
Treatment of Jurkat T cells with liquiritigenin blocks T cell activation. (**A**,**B**) Jurkat T cells (5 × 10^5^/well in 24-well plate) were pre-treated with the indicated concentration of liquiritigenin (left panel), or 40 μM of liquiritigenin (right), for 1 h. For (**A**), cells were stimulated with 100 nM of PMA and 1 μM of A23187 for 6 h (left), or the indicated time (right). For (**B**), cells were seeded on the plate pre-coated with 10 μg/mL of anti-CD3 antibodies and then added with soluble anti-CD28 antibodies (3 μg/mL) for 6 h (left) or the indicated time (right). mRNA level of IL-2 was determined by conventional PCR (top) and quantitative real-time PCR (bottom). (**C**) Stimulated Jurkat T cells with PMA/A23187 (left) or anti-CD3/CD28 antibodies for 24 h were removed, and supernatants were collected for further assay. Human IL-2 from supernatants was measured by ELISA assay. (**D**) After stimulation of Jurkat T cells with PMA/A23187 (left) or anti-CD3/CD28 antibodies for 16 h, cells were stained with anti-CD69 antibodies conjugated with APC. Cells were acquired by flow cytometry. Results are expressed as mean ± SEM of three independent experiments (* *p* < 0.05).

**Figure 4 biomolecules-10-00786-f004:**
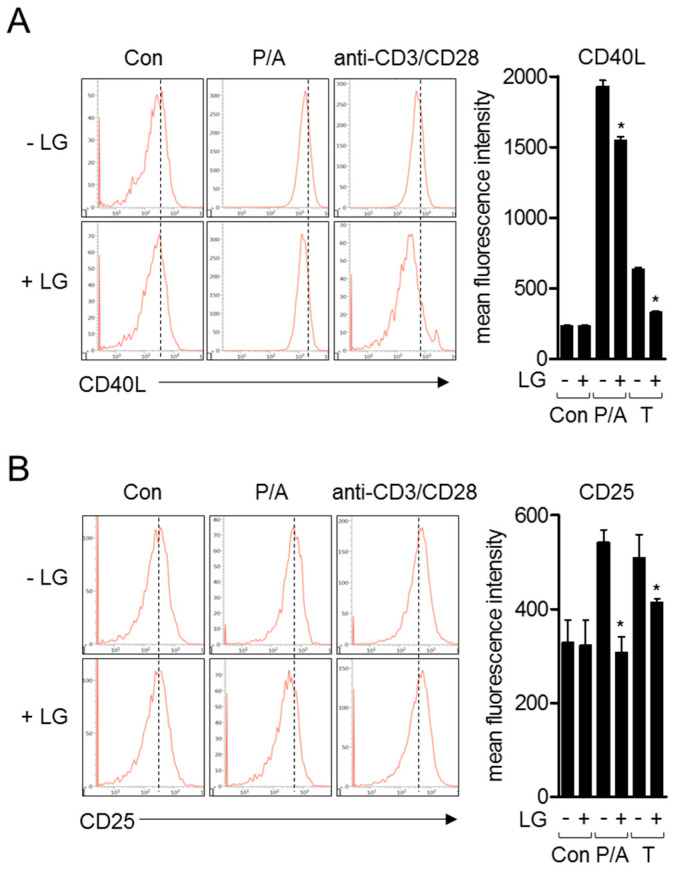
Liquiritigenin reduces the expression of surface molecules on activated T cells. (**A**,**B**) Jurkat T cells (5 × 10^5^/well in 24-well plate) were pre-treated with 40 μM of liquiritigenin for 1 h. Cells were stimulated with PMA/A23187 or coated plates with antibodies against CD3 and soluble anti-CD28 antibodies for 24 h. After collection, cells were stained with the indicated antibodies conjugated with APC (CD40L and CD25) for 1 h at 4 °C. Cells were acquired by flow cytometry for obtaining histograms, as shown. Results are expressed as mean ± SEM of three independent experiments (* *p* < 0.05).

**Figure 5 biomolecules-10-00786-f005:**
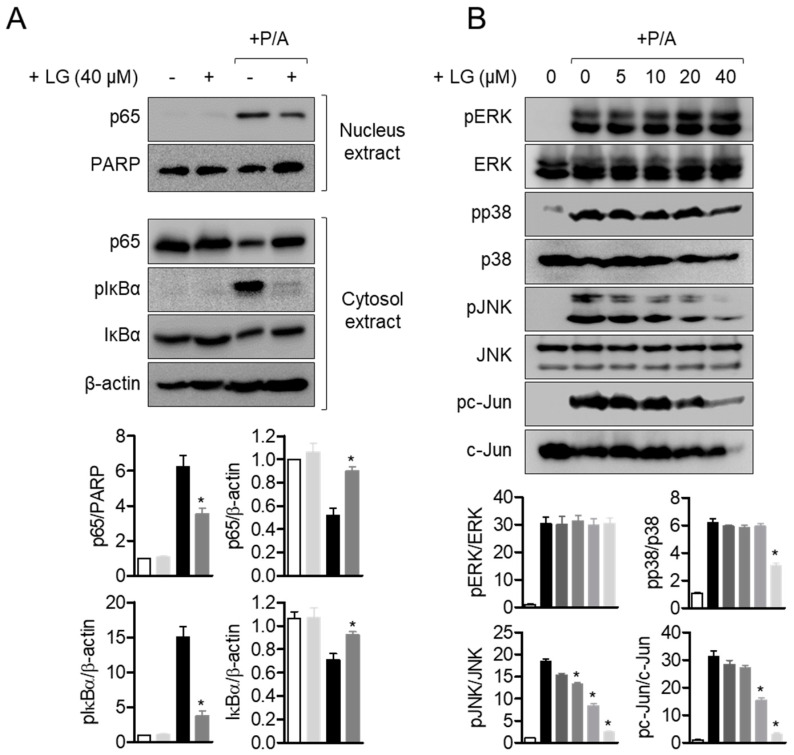
Liquiritigenin inhibits p65 translocation and MAPK signaling pathways in activated T cells. (**A**) Starved Jurkat T cells in RPMI plain media for overnight were pre-treated in the presence or absence of 40 μM of liquiritigenin for 1 h. After cells were stimulated with PMA/A23187 for 1 h, cells were immediately harvested for Western blot analysis. For separation of nucleus extract from whole lysates, NE-PER kit was used. Nucleus protein was normalized by the expression of PARP, and cytosol proteins were normalized by the expression of β-actin. (**B**) Starved Jurkat T cells were pre-treated with the indicated concentration of liquiritigenin for 1 h. Cells were stimulated with PMA/A23187 for 30 min, and immediately collected for Western blot analysis. Phosphorylation levels of indicated proteins were normalized with each total protein. All blots were presented as representative among three independent experiments in top panel, and quantified graphs were shown in bottom panel. Results are expressed as mean ± SEM of three independent experiments (* *p* < 0.05).

**Figure 6 biomolecules-10-00786-f006:**
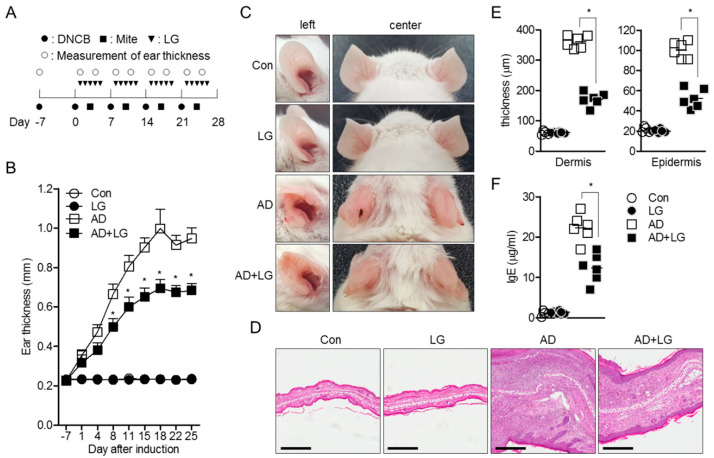
Oral administration of liquiritigenin mitigates atopic dermatitis in mice. (**A**) Experimental scheme of AD induction using DNCB and mite extract. DNCB (●) and mite extract (■) were alternatively painted weekly for 4 weeks for AD induction. Liquiritigenin (▼) of 1 mg/mL was orally administrated 5 times a week for 4 weeks. Ear thickness (○) was measured after topical administration of DNCB and mite extract. (**B**) Progress of ear thickness during 35 days. The value indicates an average of right ears and left ears from six mice. (**C**) Representative pictures of mice from each group on day 28. (**D**) Representative pictures of left ears from each group after H&E staining on day 28. Black bar indicates 300 μm. (**E**) Thickness of dermis and epidermis from images of H&E staining on day 28. (**F**) IgE level from blood serum from each group on day 28. Con, untouched mice; LG, control mice orally administrated with liquiritigenin; AD, control AD mice induced by DNCB/mite extract; AD+LG, experimental AD mice induced by DNCB/mite extract, and simultaneously orally administration of liquiritigenin. Results are expressed as mean ± SEM of six mice in same group (* *p* < 0.05).

**Figure 7 biomolecules-10-00786-f007:**
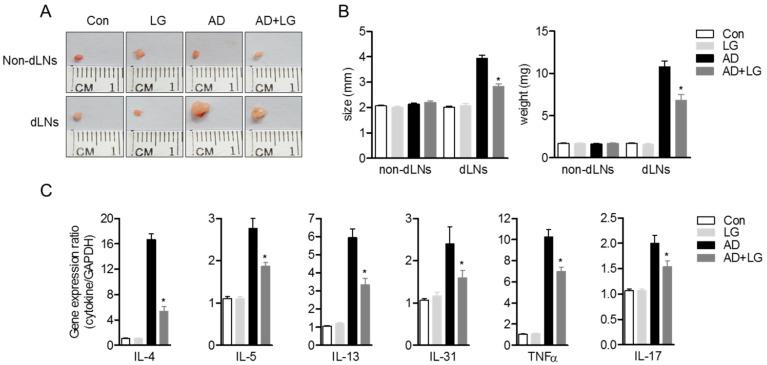
Oral administration of liquiritigenin on AD mice systemically decreases the expression of effector cytokines from effector T cells. (**A**) Representative pictures of non-dLNs and dLNs from each group on day 28. (**B**) Size (left) and weight (right) of non-dLNs and dLNs from each group on day 28. (**C**) dLNs were removed from each mouse, and CD4^+^ T cells were isolated by MACS. mRNA levels of effector cytokines were measured by quantitative real-time PCR. Results are expressed as mean ± SEM of six mice in the same group (* *p* < 0.05).

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
