# Peer review of "Oral Administration of Liquiritigenin Confers Protection from Atopic Dermatitis through the Inhibition of T Cell Activation"

_biomolecules, 2020, doi:10.3390/biom10050786_

Round 1
Reviewer 1 Report
The manuscript “Oral administration of liquiritigenin confers protection from atopic dermatitis through the inhibition of T cell activation” told us an interesting story, but some question are as follows.
- Why did you choose the dosage in oral administration of liquiritigenin?
- In order to testify the TLRF/NF-KB is involved the signal pathway, why did not use TLR4-antibody to block the TLR4 expression?
Reviewer 2 Report
Comments,
The manuscript concerns the oral administration of liquiritigenin confers protection from atopic dermatitis through the inhibition of T cell activation. The authors were studied further biological studies for natural flavanone liquiritigenin, which was isolated in 2015 from glycyrrhiza uralensis and possess strong estrogenic properties. Present manuscript authors showed that the liquiritigenin has an anti-atopic effect via control of T cell activation, and exhibits therapeutic potential for T cell-mediated disorders blocks T cell activation without cytotoxicity. These results are impressive and overall the manuscript is written well. therefore the manuscript could be acceptable for Biomolecules.
The reference for Nf-kB inhibitions is missing, include below reference for Nf-kB inhibition
- Mohapatra, D. K.; Reddy, D. S.; Ramaiah, M. J.; Ghosh, S.; Pothula, V.; Lunavath, S.; Thomas, S.; Valli. S. N.; Bhadra, M. P.; Yadav, J. S. Rugulactone derivatives act as inhibitors of NF-κB activation and modulates the transcription of NF-κB dependent genes in MDA-MB-231cells. Bioorg. Med. Chem. Lett Bioorg. Med. Chem. Lett. 2014, 24,1389-1396.
- Mohapatra, D. K.; Das, P. P.; Reddy, D. S.; Yadav, J. S. First total syntheses and absolute configuration of rugulactone and 6 (R)-(4′-oxopent-2′-enyl)-5, 6-dihydro-2H-pyran-2-one. Tetrahedron Lett. 2009, 50, 5941-5944
